# Design and Experiment of Online Detection System for Water Content of Fresh Tea Leaves after Harvesting Based on Near Infra-Red Spectroscopy

**DOI:** 10.3390/s23020666

**Published:** 2023-01-06

**Authors:** Shishun Wang, Zhengmin Wu, Chengmao Cao, Minhui An, Kun Luo, Liang Sun, Xiaoran Wang

**Affiliations:** 1School of Engineering, Anhui Agricultural University, Hefei 230036, China; 2School of Tea and Food Science, Anhui Agricultural University, Hefei 230036, China

**Keywords:** near-infrared spectroscopy, water content, tea fresh leaf, online detection system

## Abstract

Fresh tea leaves continuously lose water after harvesting, and the level of water content directly affects the configuration of tea processing parameters. To address this problem, this study established an online detection system for the water content of fresh tea leaves after harvesting based on near-infrared spectroscopy. The online acquisition and analysis system of the temperature and humidity sensor signal data was developed based on LabVIEW and Python software platforms. Near-infrared spectral data, environmental temperature, and humidity were collected from fresh leaves after harvesting. Spectral data were combined with PLS (partial least squares) to develop a prediction model for the water content of fresh tea leaves. Simultaneously, data communication between LabVIEW and PLC was established, laying the foundation for establishing a feedback mechanism to send the prediction results to the main platform of the lower computer. This provides a more objective and accurate basis for the detection of fresh leaves before processing and regulation during processing, thereby effectively promoting the standardisation and intelligent development of tea-processing equipment.

## 1. Introduction

After harvesting, the physical and chemical properties of the fresh tea leaves will continue to change. One of the main influences is the moisture content of fresh tea leaves. The water content of fresh tea leaves affects the setting of the processing parameters during the killing and kneading stages of the process. After the fresh tea leaves are killed, the water content should be between 58% and 64%. A water content that is significantly high or low is not conducive to the subsequent shaping of finished tea [1]. In the kneading process after killing, the kneading machine should decide the amount of kneading pressure and kneading time according to the water content of the killed leaves. Therefore, it is necessary to set different processing parameters depending on the water content of different harvesting batches. In most small tea processing plants in China, the water content of fresh tea leaves relies on the sensory experience of workers for testing. Therefore, the obtained test results are insufficiently rigorous. When the tea factory is in the peak production season, the water content of fresh tea leaves predicted by wokers usually have a greater deviation from the real water content of fresh tea leaves.

In recent years, non-destructive testing technology, with its non-destructive, rapid, and other advantages in the field of food testing in China, has rapidly developed. Commonly used non-destructive testing techniques include spectroscopy, machine vision [2,3,4], electronic nose [5,6,7], dielectric properties [8,9,10], nuclear magnetism [11,12,13], and acoustics [14,15]. Near-infrared spectroscopy has been widely adopted owing to its high sampling flexibility, speed, low cost, and non-destructive performance of the detected samples [16,17,18]. Currently, near-infrared spectroscopy has proven its effectiveness in qualitative and quantitative analyses in different fields, such as agriculture, food, chemical, and petroleum industries [19,20,21,22]. Wang Shengpeng et al. [23] used the partial least squares (PLS) method to establish a rapid prediction model for water, crude fibre, and total nitrogen content of fresh tea leaves, and then derived the first method to calculate the quality index (QI) of fresh tea leaf raw materials; the larger the QI, the better the quality of fresh leaves. Dong Chunwang et al. [24] successfully identified black teas with different fermentation degrees by screening the near-infrared spectral data for different fermentation degrees using PLS and the randomised frog-hopping algorithm to establish a support vector regression model. Chen et al. [25] used near-infrared spectroscopy to achieve rapid identification of different types of teas, such as green, black, and oolong teas. Three types of teas were recognised under different parameters, with recognition rates of up to 90%, 100%, and 93.33% during training. Wei Lingdong et al. [26] used near-infrared spectroscopy to discriminate the nature and taxonomic characteristics of all black teas using 200 black tea samples as the study object, combined with chemometric methods, and then by quantitative analysis, with a discrimination rate of 78.5%. Firmani et al. [27] used near-infrared spectroscopy combined with partial least squares discriminant analysis to identify darjeeling black tea, ceylon black tea, and English morning tea; among the 140 samples, 138 samples were correctly classified with an accuracy of 98.57%. Hall et al. [28] confirmed that near-infrared spectroscopy can be used for tea grade evaluation by correlating 134 black tea samples from different regions and qualities with near-infrared spectroscopy. An et al. [29] used near-infrared spectroscopy to successfully distinguish walnut kernels from walnut shells, advancing the rapid development of the walnut processing industry. From the above studies, it can be observed that near-infrared spectroscopy can classify and identify substances that can be detected using the differences in their inclusions, and the accuracy of identification and classification is high. Furthermore, this verifies that the near infrared spectroscopy detection technology is more mature in actual production. However, in the field of near-infrared spectroscopy, there is a lack of research on the online rapid non-destructive detection of the water content of tea leaves using spectroscopy techniques; therefore, this study investigates this aspect.

In this study, we developed an online detection system for the water content of fresh tea leaf samples after harvesting, based on near-infrared spectroscopy combined with the PLS algorithm to model the water content of fresh tea leaf samples. The system can quickly and accurately detect the water content, and successfully establish communication with the lower computer, laying the foundation for the lower computer to regulate the process parameters of fresh tea leaves. The design of this system significantly promotes intelligent and industrialised development in the field of online detection of fresh tea leaves, and can effectively increase tea processing.

## 2. Overall Design of Online Detection System

The overall design of the system is shown in Figure 1. It consists of feed, detection, data processing, and feedback control modules. The feeding module contains a tea guide with a vibrator. It is used to spread fresh tea leaves to be tested evenly and flatly on conveyor belts. The detection module includes a temperature and humidity sensor (KS-SHTE13T, Ningbo Keshun Instrument Co., Ltd., Ningbo, China), near-infrared spectrometer (NIR-M-R2, Shenzhen Puyan Internet Tecnology Co., Ltd., Shenzhen, China), and data acquisition card. It is employed to monitor the temperature and humidity of the testing environment, and to obtain spectral information of fresh tea leaves. The data-processing module consists of laptop computers, and is applied to process all types of data collected by the detection module and to detect the water content of fresh tea leaves. The feedback control module consists of a PLC. When the master control platform receives a signal sent by the host computer, the PLC executes different action instructions for it to adjust the machining process parameters.

## 3. Design of Real-Time Detection of Water Content of Tea Fresh Leaves after Harvesting

### 3.1. Principle of Real-Time Detection of Water Content of Tea Fresh Leaves

Near-infrared spectra are electromagnetic waves with wavelengths in the range of 780 –2526 nm [30]. The main spectral information in the near-infrared region is derived from the frequency doubling, combined frequency absorption characteristics of hydrogen-containing groups, such as C-H, N-H, and O-H [31]. The difference in water content of fresh tea leaves is manifested by the difference in the number of O-H groups; thus, the degree of absorption and reflection of light is different. Therefore, it is possible to predict the water content of fresh tea leaves using this principle. Figure 2 below shows the appearance of the fresh tea leaves sample. The absorbance of different wavelengths of near-infrared light corresponding to fresh tea leaves with different water contents is shown in Figure 3. In this system, the water content of fresh tea leaves is predicted using a near-infrared spectrometer of 900–1700 nm to obtain the absorbance information of fresh tea leaves. This was combined with the established prediction model to achieve rapid determination of the water content of fresh tea leaves.

### 3.2. Spectral Data Acquisition

The detection window of the spectrometer should fit with fresh tea leaves to be detected when the spectra are scanned. The movement of fresh tea leaves on the conveyor belt is obstructed if the spectrometer is fixed on the conveyor belt. Therefore, a simple spectral lifting device based on the Arduino control was designed. The spectral data acquisition platform consists of an Arduino Uno development board, T600 driver, stepper motor, and other hardware. The stepper motor forward and reverse program is written in Arduino, and then the T600 driver controls the stepper motor forward and reverse. When the motor was in forward transmission, the spectrometer was lowered to the conveyor belt for spectral data acquisition. At the end of the forward rotation program, the motor was delayed by 1 s before reversing to raise the spectrometer. A delay of 1 s was the time required for the spectrometer to perform one spectral scan. The design of the spectral data acquisition platform is shown in Figure 4.

### 3.3. Establishment of Water Content Prediction Model

In this study, the PLS model was used to predict the water content of fresh tea leaves. PLS is excellent when targeting more variables than the samples [32,33]. In this study, there were 228 spectral data points for one fresh leaf sample; thus, the choice of this method was appropriate. The principle of this method is to first determine the Latent Variable in the spectrum, rank the Latent Variable according to its cumulative contribution, and then use the score value of the Latent Variable to build a prediction model.

The main equipment used to build the prediction model test included 900–1700 nm ner-infrared spectrometer, constant temperature drying box, and one ten-thousandth electronic balance. A total of 153 fresh leaf samples were taken for the experiment, where the spectral data of each fresh tea leaf was used as the X variable; the water content corresponding to each spectrum was used as the Y variable. A total of 153 fresh tea leaf samples were collected. After each fresh tea leaf was weighed, spectra were collected for each fresh tea leaf sample. Spectral data were collected four times for each sample, and the average spectral data were assumed as variable X. The water content Y of the samples was measured according to the national standard GB8304-87 weighing method. The stability of the model was further improved by dividing all samples into training and prediction sets at a ratio of 7:3. There were no duplicate samples in the training or prediction set.

Owing to the influence of the instrument itself or the external natural environment, the acquired spectral data may be affected by noise and the baseline offset, which are irrelevant to the establishment of the prediction model. Therefore, this experiment will pre-process the raw spectral data before the prediction model is established to eliminate these effects and improve the model prediction accuracy. Common pre-processing methods include smooth, standard normal variate (snv), and multiplicative scatter correction (msc).

The model evaluation metrics included the coefficient of determination (*R*^2^), root mean square error of prediction (*RMSEP*), and *RPD*. The calculation formula is shown in Equation (1). When *R*^2^ is closer to 1, the smaller the *RMSEP* and *RPD* ≥ 2.5, the better the model performance.
(1)$$\left\{ \begin{array}{l} R^{2}=1-\frac{\sum {\left(y_{i}-f_{i}\right)}^{2}}{\sum {\left(y_{i}-\hat{y}\right)}^{2}}\\ RMSEP=\sqrt{\frac{\sum {\left(f_{i}-y_{i}\right)}^{2}}{m}}\\ RPD=\frac{1}{\sqrt{1-R^{2}}} \end{array}\right.$$
where y is the sample true value, f is the sample predicted value, $\hat{y}$ is the mean of the true value, and m is the number of samples in the prediction set.

Figure 5 shows the correlation analysis of the original spectra with the prediction model after applying the three pre-processing methods of smoothing, variable normalisation, and multiple scattering correction.

As can be observed from Figure 5, the difference between the evaluation indices after pre-processing does not change significantly after the rest of the original spectral data. Except for its *R*^2^, *RMSEP* and *RPD* values changed significantly after smoothing. In particular, the pre-processing itself increases the time of the whole prediction process, after considering the evaluation indexes and calculation efficiency. Therefore, we decided to employ a prediction model established using the original spectral data.

### 3.4. Tea Fresh Leaf Water Content Real-Time Detection Process

After the system workflow starts, the fresh tea leaves to be measured is placed into the vibrating tea guide plate, and the fresh leaves is evenly laid flat on the conveyor belt. After the spectral detection link, when the spectrometer obtains the spectrum of fresh tea leaves, the spectral data of fresh tea leaves are sent to the model established in Python. After water content has been detected, the data will sent to the lower main station control platform for parameter regulation, which is a complete online monitoring process.

### 3.5. Water Content Testing Platform Temperature and Humidity Parameters Monitoring

To ensure the accuracy of tea fresh leaf water content detection, it is necessary to reduce the impact of external environmental factors on tea fresh leaf water content detection. This system uses an analogue output temperature and humidity sensor, a sensor, and an NI data acquisition card connected to the tea fresh leaf water content detection platform for real-time environmental monitoring. A relatively new temperature collection device is a high precision temperature compensated capacitive sensor made of quartz crystal [34]. And considering that this design serves most small and medium-sized tea factories in China, the price of the equipment and the ease of operation need to be considered. Therefore, a combination of temperature and humidity sensor is chosen for this design. This sensor works by outputting the collected environmental parameters as a voltage signal, and then convert them according to the range of the sensor to derive the temperature and humidity parameters at the time of monitoring. They are shown in the LabVIEW upper computer interface, to ensure that they can intervene in time when the temperature and humidity fluctuation range is large, and reduce the interference of external factors.

## 4. Online Control of Process Parameters

The main processes in tea processing include killing, kneading, and drying. The purpose of this system was to predict the water content of fresh tea leaves from different harvesting batches before the production and processing of fresh tea leaves. Paving the way for parameter control of the subsequent process. The execution parts and execution method in the system are shown in the overall block diagram of the system in Figure 6.

### 4.1. Control Method and Hardware Implementation

The hardware control system consisted of a PLC, Arduino UNO development board, computer, stepper motor, DC motor, power supply, T600 driver, NI data acquisition card, and electrical control cabinet. The Arduino UNO development board, stepper motor, driver, and other hardware were combined to obtain the spectral data of the fresh tea leaves. The PLC is the main controller for regulating the processing parameters, and regulates the relevant processing parameters after the water content detection link. LabVIEW is used as the upper computer control software of the system, which monitors the temperature and humidity parameters of the environment as well as establishes communication with Arduino and PLC. Communication with the Arduino is used to control the periodic lifting and lowering of the spectrometer detection mechanism in the visualisation interface. Communication with the PLC sends different feedback values corresponding to different water contents to the PLC through the data acquisition card to regulate the processing parameters.

While the computer communicates with the lower computer, communication with Python is established through the Python node in the library function in LabVIEW. The collected spectral data were sent to the trained prediction model in Python in the form of an array to predict the water content of fresh tea leaves. Python then sends the predicted values back to the host computer, which sends appropriate commands to the PLC based on the moisture content of the batch of fresh leaves. After the signal is received, the PLC executes the corresponding program to regulate and control each process parameter.

### 4.2. Control Method and Software Implementation

#### 4.2.1. Design and Implementation of the Upper Computer Control Program

To visualise the entire online inspection system, a set of interfaces for the upper computer inspection system was designed based on the LabVIEW software. At this interface, the water content of fresh tea leaves and temperature and humidity of the tea fresh leaves to be tested environment can be clearly observed, which helps to show all the information of the entire system more intuitively, and provides a convenient adjustment to the testing system. Figure 7 shows the interface of the online inspection system of the upper computer.

#### 4.2.2. Establishment of Communication between the Host Computer and Master Station

After the prediction of the water content and OPC communication between the host PC and master PLC, the host PC discriminates the predicted water content. The result is sent to the PLC via OPC, and then the PLC regulates the following parameters: killing machine speed, kneading machine speed, level of kneading machine pressurisation, and number of kneading machine pressurisations. OPC communication of the PLC should be established with PC Access SMART software, and then data binding of corresponding controls in Labview can realise the communication between Labview and PLC.

## 5. System Testing

### 5.1. Control Method and Software Implementation

Twenty-seven samples of fresh tea leaves of the same variety were randomly collected from the tea fields. The spectral data of the 27 samples with the corresponding water contents were obtained according to the experimental method used to build the prediction model. The correlation analysis between the true water content and predicted water content is shown in Figure 8. After calculation, we found that the average absolute value of the error between the true and predicted values of the 27 samples was 0.62%, *R*^2^ was 0.8981, and *RMSEP* was 0.5119. The prediction accuracy is in line with the expected results; therefore, the application of the spectrometer enables fast and non-destructive detection of the water content of fresh leaves after harvesting on the production line. This proves that the chosen classification model, with the use of a spectrometer to detect the water content, is more effective, providing technical support for the ultimate goal of quickly adjusting the processing parameters according to the water content.

### 5.2. Control Method and Software Implementation

The following experimental protocol was designed to verify the smooth operation of the online detection system: After considering samples from the tea field, the samples were divided into three portions, and each sample of fresh tea leaves was spread evenly on the conveyor belt of the testing system. Subsequently, the testing system was activated, and the average water content of each fresh tea leaf was predicted by averaging the predicted values of water content at several sampling points of spectral testing in each fresh tea leaf. Under the condition that the time required to complete an inspection by the spectrometer is constant, the number of spectral inspections will vary by setting different conveyor running speeds. The purpose of this experiment was to investigate the relationship between the running speed of the conveyor belt and the prediction accuracy of the water content, as well as the relationship between the number of spectrometer tests and the prediction accuracy of the average water content of each tea leaf.

After setting, the time required to collect the spectral data was 2 s. The total length of the conveyor belt was 60 cm, and belt speeds of 2, 3, and 6 cm/s were set for this experiment. The corresponding numbers of spectral detections were 15, 10, and 5.

From Figure 9 and Table 1, it can be observed that the prediction accuracy of fresh tea leaf water content decreases continuously with an increase in conveyor operation speed. An increase in the number of tests per sample by the spectrometer can effectively improve the prediction accuracy of the average water content of each batch of fresh tea leaf samples. The prediction of the whole machine is better when the running speed of the conveyor is 3 cm/s, considering the actual productivity.

## 6. Conclusions

A near-infrared spectrometer-based online detection system for the water content of tea leaves after harvesting was designed, which can detect the water content of fresh tea leaves before processing and monitor the temperature and humidity of the detection environment. The model performance experiments showed that the *R*^2^, *RMSE*, and *RPD* values all met expectations; the average error between the true and predicted values was only 0.62%, which was a good prediction, and met the demand for rapid non-destructive testing.

In the case of the whole machine experiment, when the conveyor speed was 3 cm/s, the error of the prediction result was only 0.69%. The results do not differ significantly from the experimental error of the model performance, which verifies the feasibility of online inspection, and provides data support for the subsequent regulation of machining process parameters.

## Figures and Tables

**Figure 1 sensors-23-00666-f001:**
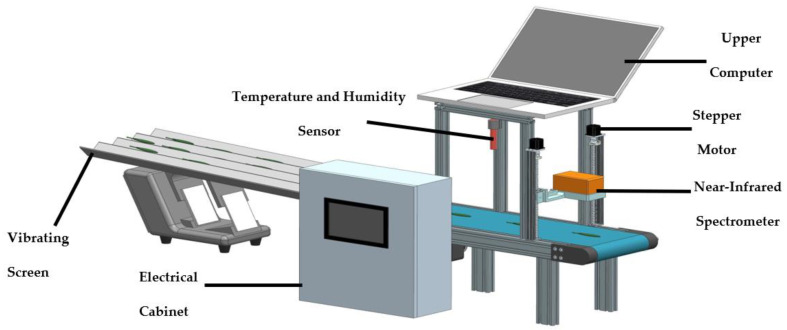
Online monitoring system overall design diagram.

**Figure 2 sensors-23-00666-f002:**
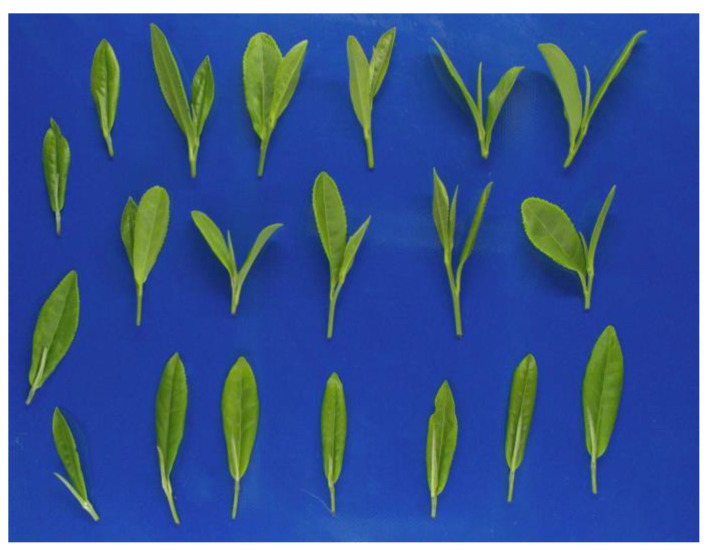
Appearance of the fresh tea leaves.

**Figure 3 sensors-23-00666-f003:**
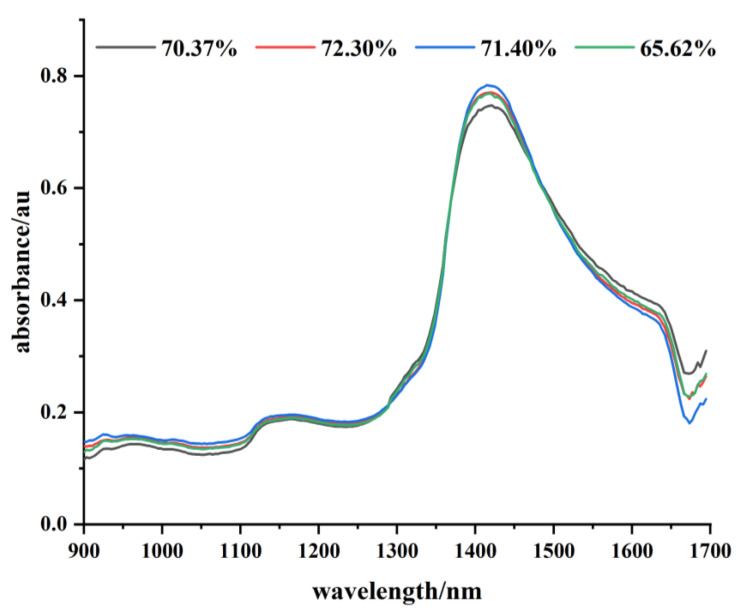
Spectral images of tea leaves with different water contents.

**Figure 4 sensors-23-00666-f004:**
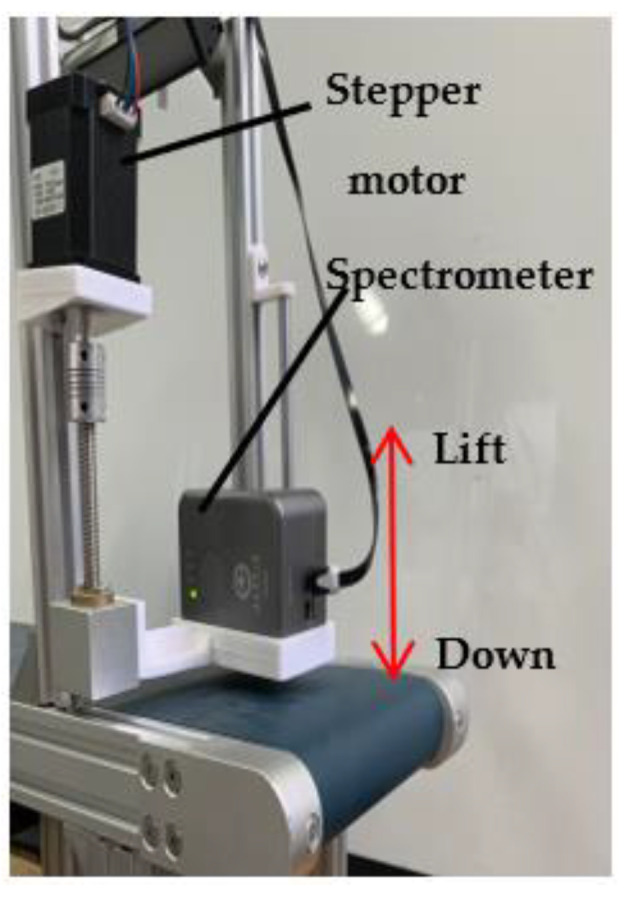
Design of spectral data acquisition platform.

**Figure 5 sensors-23-00666-f005:**
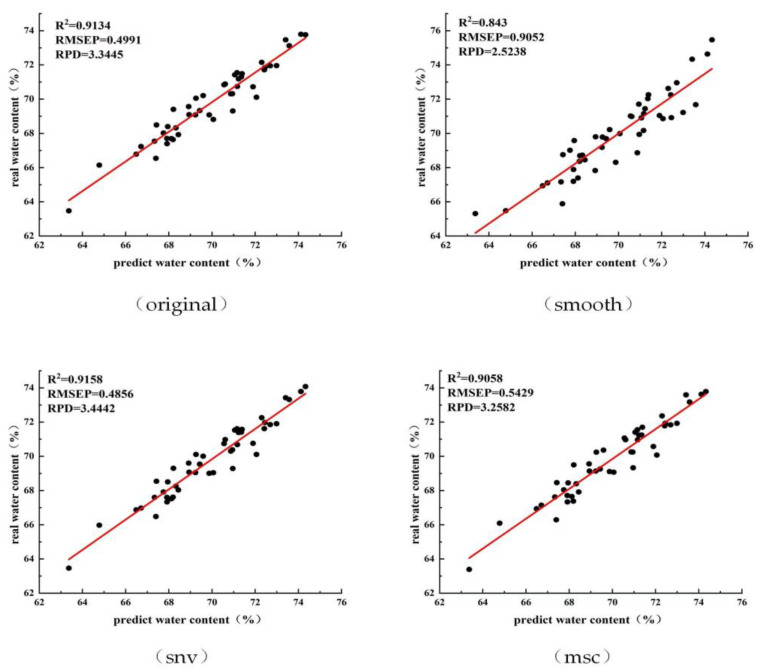
Correlation analysis of the effect of different pre-processing methods on prediction accuracy.

**Figure 6 sensors-23-00666-f006:**
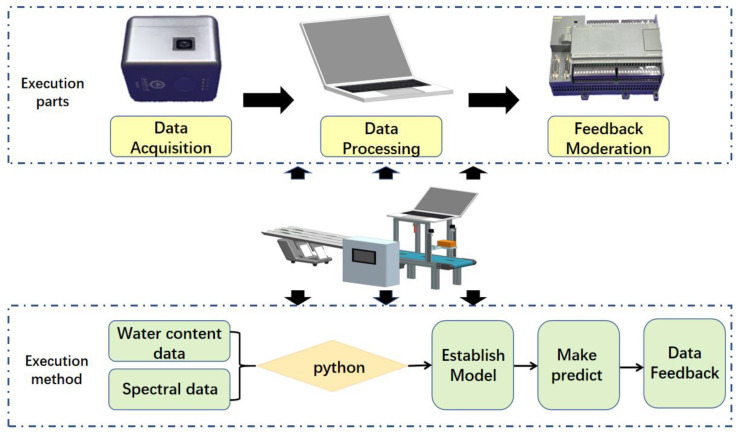
Overall system block diagram.

**Figure 7 sensors-23-00666-f007:**
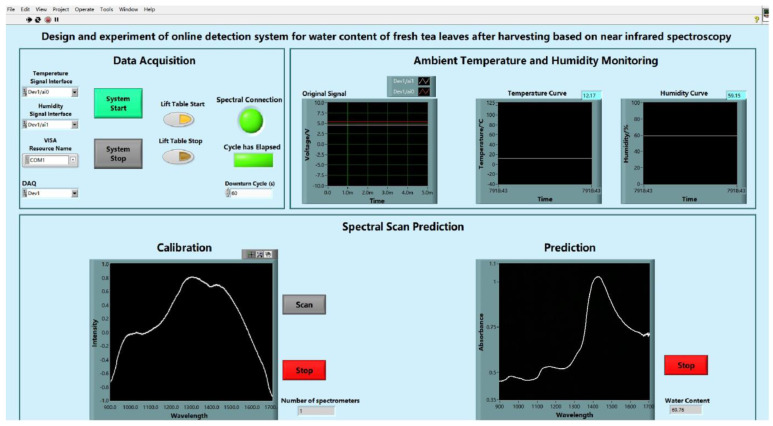
Labview upper computer online detection system interface.

**Figure 8 sensors-23-00666-f008:**
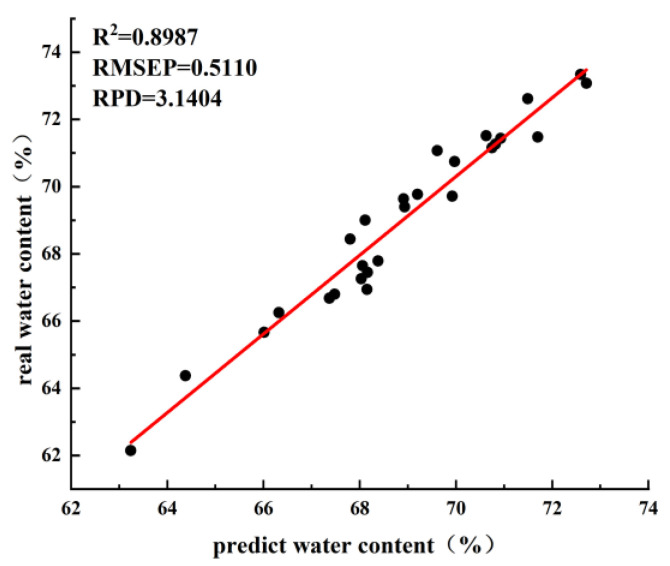
Correlation analysis of real and predicted values of water content of fresh leaf samples.

**Figure 9 sensors-23-00666-f009:**
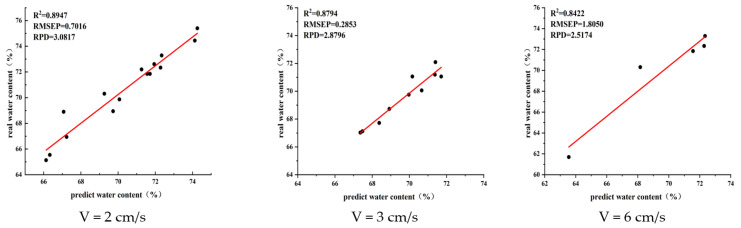
Analysis of correlation between conveyor operation speed and prediction accuracy.

**Table 1 sensors-23-00666-t001:** Comparison of the number of spectral tests and the prediction accuracy of the average water content of fresh leaves.

Testing Times	Average Predicted Value	Average Overall Value	Error Absolute Value
5	69.89%	71.64%	1.75%
10	69.58%	70.27%	0.69%
15	70.64%	70.16%	0.48%

## Data Availability

No applicable.

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
