# Peer review of "Design and Experiment of Online Detection System for Water Content of Fresh Tea Leaves after Harvesting Based on Near Infra-Red Spectroscopy"

_sensors, 2023, doi:10.3390/s23020666_

Round 1
Reviewer 1 Report
The article is not written precisely enough.
Page 2: 1. Introduction: line 43: There are also highly sensitive methods for measuring humidity or dielectricity. These methods are temperature compensated and use highly sensitive quartzes as described in:
-Temperature-compensated capacitance-frequency converter with high resolution. Sensors and actuators. A, Physical, ISSN 0924-4247, 2014, vol. 220, p. 262-269, doi: 10.1016/j.sna.2014.09.022.
Authors should mention these references above in the text.
Page 3: line 87: Which humidity sensor? Name it!
Page 3: line 111: What do these tea leaves even look like? How do shape, volume, and size affect the results in Figure 2?
Page 5: line 148: Why is weighing even used if all tea leaves are different shapes and have different volumes.
Page 7: line 215: Which spectrometer? Name it!
Page 8: Figure 6: Where are diagrams of measurement?
Reviewer 2 Report
The author presented A near-infrared spectrometer-based online detection system for the water content of tea leaves after harvesting was designed, which can detect the water content of fresh tea leaves before processing and monitor the temperature and humidity of the detection environment. The presentation, objective and significance of study is clear, overall structure of manuscript is well managed, however few changes are required to accept for publication.
Author should provide references to statements from line 26-39. e.g“The water content of fresh tea leaves affects the setting of the processing parameters during the killing and kneading stages of the process. After the fresh tea leaves are killed, the water content should be between 58% and 64%” and “ A water content that is significantly 30 high or low is not conducive to the subsequent shaping of finished tea. In the kneading 31 process after killing, the kneading machine should decide the amount of kneading pres- 32 sure and kneading time according to the water content of the killed leaves. Therefore, it is necessary to set different processing parameters depending on the water content of different harvesting batches” other wise these statments need explaination interms of results.
Line 49 "prediction model for water??" is it water content of water level
In Figure 1 ,3 it would be better to write the original text rather than number
Figure 2 figure is not clear should provide high resolution image
Line 134 PLS Model mathematical interpretation is not provided is will be useful to provide it or gave a reference
Line 161 it would be better to paste figure 4 and its description after the 177 after equation.
It would be better to provide quality reference studies eg. [14-19][23][26]
